# The Effects of Gender-Affirming Hormone Therapy on Quality of Life: The Importance of Research on Youth

**DOI:** 10.3390/healthcare12131336

**Published:** 2024-07-04

**Authors:** Monika Mazur, Paweł Larionow

**Affiliations:** 1Independent Researcher, 2500 Valby, Denmark; 2Faculty of Psychology, Kazimierz Wielki University, 85-064 Bydgoszcz, Poland

**Keywords:** gender, gender affirmation, gender-affirming hormone therapy, non-binary, psychological well-being, sex, transgender, transgender female, transgender male, transgender youth

## Abstract

Gender-affirming hormone therapy (GAHT) plays a significant role in the medical care of transgender individuals, helping to align their physical characteristics with their gender identity. While numerous studies have investigated the impact of GAHT on adults, research focusing on its effects on the quality of life (QoL) of transgender youth is limited. In this opinion paper, we aim to address selected challenges associated with gender-affirming medical care, such as (1) the necessity for evidence-based youth gender-affirming medical care, (2) the urge to explore different approaches to gender-affirming medical care diversely in transgender youth research, and (3) understanding the challenges of the detransition process (which refers to stopping or reversing gender-affirming medical or surgical treatments), as well as suggest possible solutions for meaningful progress. Notably, the available evidence underlines a positive impact of GAHT on various aspects of QoL of transgender youth, such as mental health and social functioning, by alleviating gender dysphoria, improving body satisfaction, and facilitating appearance congruence (the degree to which an individual’s physical appearance represents their gender identity). However, challenges related to methodological limitations, as well as ethical considerations, and several sociocultural factors highlight the need for further research to better understand the long-term effects of GAHT on the QoL of transgender youth. Ethical considerations, such as ensuring informed consent and weighing potential benefits against risks, are pivotal in guiding healthcare decisions. Additionally, navigating these ethical responsibilities amid sociocultural contexts is crucial for providing inclusive and respectful care to transgender youth. Addressing these research gaps is, therefore, crucial to developing successful healthcare programmes, raising awareness, and promoting the holistic well-being of transgender youth through comprehensive and affirming care.

## 1. Introduction

Transgender and gender diverse are the umbrella terms for people whose gender identities or expressions differ from cultural expectations and norms associated with the sex they were assigned to at birth [1]. Although gender diversity is not a recent phenomenon, transgender and gender-diverse individuals are frequently exposed to stigmatisation, transphobia, and minority stress [1]. These factors, in some cases, can lead to significant psychological distress [1]. Furthermore, some transgender and gender diverse people can experience gender dysphoria, which some researchers define as “distress due to a discrepancy between one’s assigned gender and gender identity” [2] (p. 1). Many transgender individuals decide to pursue the transition process to improve their well-being and to live a coherent life [1].

Current research shows that, in recent decades, the number of individuals seeking gender-affirming medical care has consistently increased [3]. The growing body of the literature indicates, among others, such possible reasons for this shift: (1) a shift in social attitude towards the transgender community (i.e., raising acceptance, destigmatisation, etc.), (2) increased awareness of transgender issues, and (3) easier access to gender-affirming medical care [3]. These factors could help young people explore and express their gender identities [4] and foster greater societal acceptance [5]. However, researchers underline that since adolescence and young adulthood are stages of dynamic identity development [5], in some cases, the transition phenomenon could arise due to social contagion [6]. Given limited high-quality evidence and the controversial character of this statement, further research is required.

Considering the evolving understanding of sexual health, the 11th edition of the *International Statistical Classification of Diseases and Related Health Problems* has rectified gender-identity-related health classification [7]. Terms like “transsexualism” and “gender identity disorder of children”, included in the 10th edition of the *International Statistical Classification of Diseases and Related Health Problems*, have been replaced with more inclusive terms like “gender incongruence of adolescence and adulthood” and “gender incongruence of childhood” [7]. This reflects a commitment to respect and value all individuals. The fifth edition of the *Diagnostic and Statistical Manual of Mental Disorders* has also updated its terminology and now defines gender dysphoria as “distress that may accompany the incongruence between one’s experienced or expressed gender and one’s assigned gender” [8] (p. 451). This is a more detailed and inclusive definition compared to the term “gender identity disorder” used in the fourth edition of *Diagnostic and Statistical Manual of Mental Disorders* [9]. The above-described changes aim to grant transgender individuals wider access to gender-affirming healthcare. Despite observed positive results concerning quality of life (QoL), access to gender-affirming healthcare remains obstructed, especially for transgender youth, due to unclear procedures, limited visibility on long-term results, and ethical dilemmas.

First, recent research demonstrates that gender-affirming hormone therapy (GAHT) could improve psychological well-being and body image satisfaction in adult transgender individuals [10]. However, more high-quality evidence is needed to confirm the influence of GAHT on their psychosocial functioning [11]. Secondly, despite the fact that available studies suggested that GAHT in transgender youth could be relevant to improving their self-perception [12], prospective evidence on the youth population is scarce [13].

Furthermore, questions revolve around gender identity, rights, and self-determination limits [14]. In some countries, changing marital status or filing a lawsuit against one’s parents/caregivers may be required to initiate the gender recognition process. These practises may be considered discriminatory and remain discussable due to the significant psychological distress caused to the concerned individuals. Moreover, healthcare professionals’ limited competencies and low awareness of transgender-specific issues can strongly affect the psychological well-being of transgender individuals, aggravating their gender dysphoria [15]. This issue can be particularly threatening to transgender youth, who are considered a vulnerable population at significantly elevated risk of suicide [16].

Ultimately, considering the above ideas, through this paper, we would like to bring to light the following topics, which we consider crucial to understanding the situation of transgender youth population:(1)Necessity for evidence-based youth gender-affirming medical care;(2)Urge to explore different approaches to gender-affirming medical care diversely in transgender youth research;(3)Ethical considerations surrounding GAHT in youth;(4)Understanding the challenges of the detransition process.

We believe that raising awareness related to challenges in transgender youth is crucial to promoting the holistic well-being of this population through comprehensive and affirming care.

## 2. The Necessity for Evidence-Based Youth Gender-Affirming Medical Care

Recent research conducted in the United States reveals a significant increase in the number of young people identifying as transgender. With over 1.6 million adults (aged 18 and older) and youth (ages 13 to 17) identifying as transgender [17], the transgender and gender-diverse community has gained prominence in media discourse. However, despite the growing recognition of their voices, constraints on the diversity of transgender and gender-diverse representation in the media persist [18].

This shift in representation has drawn attention to the challenges faced by transgender individuals, including minority stress, discrimination, and gender dysphoria, which impact their QoL and well-being [19]. Research on transgender adults demonstrates elevated risks of adverse mental health outcomes, such as depression and anxiety, often leading to substance abuse or suicidal behaviour. Cultural differences in the degree of sexual and gender diversity acceptance have an important role in this context [20]. Moreover, individuals experiencing gender incongruence report higher levels of suicidal ideation and lifetime suicide attempts compared to their cisgender peers [21,22], indicating vulnerability of the transgender population and underlining the importance of actions that should be taken to promote equality and consequently reduce discrimination.

Since transgender individuals are at five times higher risk of suicidal ideations than cisgender heterosexual adolescents [22], questions arise regarding available gender-affirming medical treatments. Current studies primarily focus on transgender adults, supporting the effectiveness of feminising or masculinising GAHT [11]. Despite its globally increasing adoption, gender-affirming therapy remains controversial. Due to limited literature and various study constraints such as small sample sizes, lack of diversity, and ethical concerns, doubts emerge regarding its suitability, safety, and effectiveness. Furthermore, some researchers discuss the implication of GAHT on reproductive capacity and highlight the importance of fertility preservation counselling prior to initiation of the relevant treatments [23]. Other research outlines the impact of GAHT on metabolic profile and an increase in blood pressure because of it [24]. Moreover, transgender youth demonstrate higher rates of overweight and obesity [25], and this underscores the importance of further research [26]. Consequently, there is a call for further high-quality studies and prospective controlled trials [27]. As research predominantly targets adults, the debate now should be extended to gender-affirming medical care for children and youth, with multiple factors being considered amidst limited evidence. Uncertainties persist regarding the long-term effects and optimal approaches to such treatments.

Recent research highlights that expressing one’s perceived gender identity (e.g., through behaviours, mannerisms, and appearance) can promote harmonious gender identity development and contribute to psychological well-being [28]. However, since gender-identity-related studies are frequently conducted on cisgender, White, and Eurocentric youth [29], the visibility of experiences in culturally diverse populations remains unclear. Consequently, additional high-quality research is necessary to obtain a deeper understanding of gender identity development in gender-diverse individuals coming from different backgrounds to satisfy their needs better. Therefore, it is crucial to prioritise further research and evidence-based approaches to gender-affirming medical care for children and youth, ensuring their access to safe and effective treatments while addressing the ongoing controversies and uncertainties surrounding these interventions.

## 3. Assessing Different Approaches in Gender-Affirming Medical Care Diversely in Transgender Youth Research

Since the main developmental task of adolescence is identity formation [30], some individuals could experience an increase in gender dysphoria following pubertal changes [31]. Healthcare practitioners strive to meet individual needs while prioritising caution and safety in managing gender-affirming care through different paradigms.

On the one hand, the traditional paradigm of “gatekeeping” refers to clinicians applying eligibility criteria (i.e., persistent and documented gender dysphoria, well-controlled medical or/and mental concerns, etc.) to determine an individual’s readiness to undergo gender-affirming care [32]. Although this approach may not recognise individual experiences of transgender individuals [33], the narrative behind it is to avoid unnecessary harm and possible regret related to transitioning [32]. The main limitations of “gatekeeping” underlined by researchers are the restricted autonomy of an individual willing to pursue transition [34] and the necessity to participate in a specific number of therapy sessions before medical interventions are approved [35]. These elements could constitute a barrier to establishing a solid therapeutic relationship based on trust [34] and, consequently, question this paradigm’s validity.

On the other hand, the eighth version of the *Standards of Care for Transgender and Gender Diverse People* (SOC-8), provided by the World Professional Association for Transgender Health (WPATH), advocates for patient-centred care through transparent communication [1]. According to the SOC-8, psychotherapy and educational interventions are recommended for prepubertal youth instead of immediate medical treatments [1]. Another component, “social transition”, allows gender-diverse children and adolescents to adopt names, pronouns, and gender expressions that align with their identity to alleviate dysphoria [36]. Moreover, the SOC-8 allows transgender and gender-diverse adolescents at Tanner Stage 2 of puberty, who have been evaluated by a mental health gender specialist, to access gender-affirming medical care [37]. However, concerns remain about accurately applying the informed consent model [38]. Researchers argue that the growing number of youth reaching out for gender-affirming medical treatment [39] could reduce the quality of mental health evaluations before treatment [38].

Ultimately, the transition process can vary depending on the individual (i.e., some teenagers desire only social transition, while others would like to pursue gender-affirming medical care). The above-described approaches are divergent, each of them possesses unique benefits and limitations. Further research is needed to explore their efficacy and validity across various situations and conditions. Considering this complexity, building a more specific approach based on the advantages of both methodologies may benefit the individuals pursuing a gender-affirming process by providing a tailored transition experience addressing their individual needs and preferences. An inclusive and affirming multidimensional, adolescent-centred approach based on open dialogue is crucial [29,40].

## 4. Ethical Considerations Surrounding GAHT in Youth

The ethical considerations (i.e., unequitable access to gender-affirming care, self-treatment, limitations of informed consent, ensuring competencies of medical practitioners, and decision autonomy) surrounding GAHT in young people are complex and encompass various factors that significantly impact treatment outcomes. We believe that the cooperation of politicians, lawyers, social workers, and psychologists, leading to decreased homophobia and stigmatisation of people representing sexual and gender minorities [41], is important in the context of ethical considerations. Hereunder, we would like to briefly outline the selected concerns.

(1)Implications of Delay in Gender Transition Processes: Mental Health and Identity Forming Considerations

Discussions spark about the extended delays within the gender transition process. Some researchers suggest that shorter delays in awaiting gender-affirming treatment could lead to better mental health outcomes [42], whereas others indicate that longer waiting times are crucial to reach a crystallised perception of one’s gender identity, particularly in younger (5–10 years old) children [43]. People awaiting medical treatments experience a variety of emotions [43]. The vulnerability of the transgender population highlights the extended delays within the gender transition process. As a result, the debate not only centres around the role of delay in the treatment process but also on the interplay of gender dysphoria and mental health. In this context, studying demographic and psychological factors related to increased discomfort during the waiting period for gender-affirming medical care would be beneficial.

(2)Balancing Autonomy—The Debate

Informed consent aims to promote patient´s autonomy. The patient and medical practitioner achieve an agreed medical decision through an ongoing communication process [44]. Gender-affirming medical care is considered a fairly new domain in medical care [45]. In this context, informed consent may be help to alleviate the ethical unease of medical practitioners [38]. However, researchers underline that insufficient understanding of the outcomes of gender-affirming treatments can limit the patient’s autonomy [38]. Therefore, developing a more nuanced approach to ensure that the principles of informed consent truly empower individuals, balancing the need for comprehensive information with the realities of medical and psychological complexities, would be beneficial.

(3)Balancing Beneficence and Nonmaleficence

The growing number of gender-diverse youth seeking medical care is pressuring institutions and practitioners to quickly evaluate and recommend treatments [38]. This situation raises concerns about the accuracy of pre-treatment evaluations for psychological conditions in gender-diverse youth (e.g., autism, eating disorders, sexual abuse, and mental health issues) and their impact on post-transition well-being and potential regret [38]. The lack of time to carefully consider these factors challenges the principles of the informed consent model. Consequently, discussions arise regarding the ethical principles of beneficence (acting in the patient’s best interest) and nonmaleficence (avoiding harm) [46].

## 5. Exploring the Challenges of Detransition

The visibility of detransition complexities remains limited due to the scarcity of studies. To address the variety of experiences within the transgender community, one of the most relevant needs is obtaining new evidence on discontinuation and detransition phenomenon within high-quality, evidence-focused GAHT [47].

First, there is a significant disparity between the visibility of detransition rates within the population and the level of media coverage dedicated to this phenomenon. The limited literature on the subject could be due to scientists’ reluctance to stigmatise transgender individuals and their perception of gender as a spectrum [47]. This attitude can be justified by ethical reasons (i.e., considering detransition a “taboo” within the transgender community). Nevertheless, the practical consequences (i.e., lack of adequate detransition care procedures due to limited knowledge) could harm the population. Thus, apart from a better understanding of the detransition phenomenon per se, it is recommended that future research investigates the discontinuation of the transition process in a more holistic manner, considering not only individual experiences but also societal, psychological, and medical factors.

Secondly, given that detransition testimonials often allude to transition regrets, a debate emerges about access to medical transition for teenagers [48]. It is important to underline that detransition could be caused by multiple reasons. Researchers outline two case scenarios for detransition occurrence. In the first case, also called “core” or “principal”, stopping and reversing the transition process is a consequence of a person no longer identifying as transgender. This shift could be caused by multiple factors such as finding a different way of coping with gender dysphoria or understanding the impact of past trauma or internalised transphobia on the experience of gender dysphoria [49]. In the second case, known as “secondary” or “non-core”, a person keeps identifying as transgender but decides to stop transitioning due to external factors such as lack of family/society support [49]. Other reasons for stopping the transition in this context could be dissatisfaction with the results or health concerns [49]. Moreover, the desire to become a parent has been identified as a factor influencing the decision to halt transitioning [49]. Overall, controlling for the adverse mental health outcomes (e.g., anxiety symptoms) in research would be helpful to reveal the unique role of mental health symptoms in the overall psychological distress of transgender people before, during, and after gender-affirming treatment. Longitudinal studies investigating improvements and trajectories in gender dysphoria could provide insights into the nature of dysphoria and whether these improvements might be linked to mental health factors. Additionally, exploring the efficacy of psychotherapy and educational interventions in alleviating suffering could be beneficial, although the underlying mechanisms are not yet fully understood.

Moreover, recent research highlights that, in some cases, external factors such as stigma and victimisation could enhance internal factors, such as depression and self-doubt regarding one’s gender identity [36]. Researchers underline that, in many cases, non-core detransitions can be temporary [49]. Furthermore, while gender identity fluctuation could occur post-treatment and result in detransition, it is essential to acknowledge that such experiences are individual and do not negate the validity of transgender identities or the importance of gender-affirming healthcare practises [47]. Yet, while not based on the results of research, discontinuation of gender-affirming medical care is often used in the anti-transgender discourse and constitutes an additional challenge to transgender individuals as a community.

Finally, despite often being considered controversial, detransition is a complex phenomenon requiring a nuanced understanding. While some of the research acknowledges that transition regret may occur (i.e., there was another problem mistaken for dysphoria) [47], the adult detransition decision is most frequently caused by adverse external factors such as family rejection, school-based harassment, and lack of support [36,47]. Research on youth detransition underscores the emotional complexity of the process, revealing coexisting feelings of satisfaction, regret, and grief [47], highlighting the critical need for adequate psychological support and understanding during this challenging period. High-quality multidimensional research across various age groups is essential to achieve a deeper awareness of detransition. Ultimately, only a comprehensive investigation focused on varied aspects of detransition, followed by the implementation of new clinical guidelines will allow us to mitigate its current divisiveness. The challenges discussed in our opinion article are summarised in Table 1, along with the possible solutions identified.

## 6. Conclusions

This opinion paper discusses controversial, multifaceted challenges surrounding gender-affirming medical care, underlining critical aspects that could be overlooked in social discourse. The identified challenges, from limited evidence on psychosocial effects to insufficient research on ethnically diverse populations, highlight the pressing need for targeted interventions. Moreover, the suggested solutions present a pathway towards meaningful progress. Emphasising evidence-based care and clarifying diagnostic classifications can lead to significant accessibility to gender-affirming healthcare, potentially enhancing the QoL of the vulnerable transgender youth population. Furthermore, the increasing visibility of transgender youth and advocating for inclusive social structures plays a pivotal role in reducing discrimination and stigmatisation, thereby granting a more supportive and inclusive environment for all. In addition, we advocate for developing a multidimensional approach that integrates the strengths of diverse methodologies to address the needs of transgender youth undergoing gender-affirming processes in the best way. We would like to believe that considering the insights and recommendations provided in this paper could result in a future where gender-affirming medical care is not only accessible but also respectful, affirming, and empowering for all individuals, regardless of their gender identity or background.

## Figures and Tables

**Table 1 healthcare-12-01336-t001:** Challenges, solutions, and outcomes related to transgender youth situation.

Challenges	Solutions	Outcomes
Limited evidence on psychosocial effects of GAHT.	To emphasise evidence-based gender-affirming care and clarify diagnostic classifications in the 11th edition of the *International Statistical Classification of Diseases and Related Health Problems*.	Increased accessibility of gender-affirming healthcare and potential improvement in QoL.
Limited evidence on long-term effects and optimal approaches to gender-affirming medical care for youth and ongoing controversies.	To recognise the growing visibility of transgender youth.	Creation of inclusive social structures as well as reductions in discrimination and stigmatisation.
Limited research on ethnically diverse populations and deficient visibility of diverse experiences in existing studies.	To adopt a multidimensional, youth-centred approach.	Improved QoL in gender-diverse youths from different cultural backgrounds.
Limited knowledge related to detransition and discontinuation of gender-affirming medical care.	To examine the detransition phenomenon in depth.	Development of a multidimensional approach supporting individuals during the overall detransition journey.
Multifaceted ethical dilemmas.	To collect data supporting the fair provision of the appropriate care and ensure equitable access to treatment.	Define and enhance informed consent, increase visibility, and provide greater autonomy in treatment-related decision-making.

## Data Availability

Not applicable.

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
