# Peer review of "The Effects of Gender-Affirming Hormone Therapy on Quality of Life: The Importance of Research on Youth"

_healthcare, 2024, doi:10.3390/healthcare12131336_

Round 1

Reviewer 1 Report

Comments and Suggestions for Authors

The abstract effectively highlights the focus on gender-affirming hormone therapy and its impact on transgender youth, which is an important and timely topic. Moreover, it identifies the limited research on the quality of life (QoL) in transgender youth and the need for evidence-based medical care, highlighting important research gaps. Finally, it covers various aspects such as mental health, social functioning, and the process of detransition, presenting a well-rounded view of the topic.

While the abstract addresses numerous challenges and aspects, it might benefit from a more focused approach to one or two main points to avoid being overly broad. Terms like "appearance congruence" and "detransition process" may require clarification at this stage.

Body

The text thoroughly addresses various aspects of gender incongruence, gender-affirming care, and related challenges, providing a well-rounded discussion. The structure is logical and clear, making it easy to follow the progression of ideas from defining gender incongruence to discussing medical care and ethical dilemmas. The text is well-supported by numerous citations from reputable sources, adding credibility to the arguments presented.

Suggestions for improvement:

1.         Some points, such as the need for further research and the benefits of gender-affirming  care, are repeated multiple times, which could be streamlined to improve conciseness.

2.         While the text mentions methodological limitations and ethical considerations, it lacks             specific examples or details that could provide deeper insight into these issues.

3.         Some sentences are long and packed with multiple ideas, which can be difficult to  digest. Breaking these into shorter, more focused sentences would improve clarity.

For example:

Lines 137 – 139:

Original: "Although its focus on possible adverse effects (i.e., fertility issues, irreversible results of gender-affirming hormone treatment) and promoting caution (i.e., in the pre-puberty stage, gender-adverse behaviors may not be due to gender dysphoria, puberty blockers can only be administrated at puberty onset) is understandable, this conventional approach might have severe unintended consequences (Vandermorris & Metzger, 2023)."

Revised: "The focus on possible adverse effects, such as fertility issues and irreversible results of gender-affirming hormone treatment, and promoting caution is understandable. This includes considering that gender-adverse behaviors in the pre-puberty stage may not be due to gender dysphoria, and puberty blockers can only be administered at puberty onset. However, this conventional approach might have severe unintended consequences (Vandermorris & Metzger, 2023)."

OR

Lines 112-114:

Original: "The existing literature underlines the importance of gender expression in the lives of transgender youth and its influence on the development of their gender identity and overall psychological health (Pullen Sansfaçon et al., 2020)."

Revised: "Existing literature highlights the importance of gender expression in the lives of transgender youth. It influences the development of their gender identity and overall psychological health (Pullen Sansfaçon et al., 2020)."

Conclusions

The conclusions effectively summarize the key points discussed in the paper, reinforcing the importance of the issues raised.They emphasize the need for targeted interventions and evidence-based care, advocating for practical changes in the healthcare system.

They advocate for an inclusive and supportive environment for transgender youth, highlighting the importance of reducing discrimination and stigmatization.

Suggestions for improvement:

·         Phrases like "potentially enhancing the QoL for the vulnerable transgender youth population" can be more assertive to strengthen the conclusion.

·         Providing a few more concrete steps or examples of how the suggested solutions can be implemented would strengthen the conclusion.

Comments on the Quality of English Language

Some sentences should be made more concise. Better break them to 2 parts and connect with a connector. See my review above, for some examples.

Author Response

Dear Editors and Reviewers,

We would like to thank the editors and the reviewers for their positive and encouraging feedback on our submission. The constructive comments of reviewers helped us to significantly improve the quality of our submission. We have been through all comments one by one, edited the manuscript in detail, and added new material where required. We hope the editor and reviewers find the revised version of the manuscript clear and suitable for publication in Healthcare. All changes made in the manuscript are highlighted in red.

Comment 1:

The abstract

Effectively highlights the focus on gender-affirming hormone therapy and its impact on transgender youth, which is an important and timely topic. Moreover, it identifies the limited research on the quality of life (QoL) in transgender youth and the need for evidence-based medical care, highlighting important research gaps. Finally, it covers various aspects such as mental health, social functioning, and the process of detransition, presenting a well-rounded view of the topic.

While the abstract addresses numerous challenges and aspects, it might benefit from a more focused approach to one or two main points to avoid being overly broad. Terms like "appearance congruence" and "detransition process" may require clarification at this stage.

Reply1: Thank you for your recommendation. Terms "appearance congruence" and "detransition process" in abstract are now clarified.

Comment 2:

Body

The text thoroughly addresses various aspects of gender incongruence, gender-affirming care, and related challenges, providing a well-rounded discussion. The structure is logical and clear, making it easy to follow the progression of ideas from defining gender incongruence to discussing medical care and ethical dilemmas. The text is well-supported by numerous citations from reputable sources, adding credibility to the arguments presented.

Suggestions for improvement:

1. Some points, such as the need for further research and the benefits of gender-affirming care, are repeated multiple times, which could be streamlined to improve conciseness.

2. While the text mentions methodological limitations and ethical considerations, it lacks specific examples or details that could provide deeper insight into these issues.

Reply 2: We now avoid repeating elements such as need for further research and the benefits of gender-affirming care to enhance reader´s experience. A paragraph discussing ethical concerns is now added.

Comment 3:

3. Some sentences are long and packed with multiple ideas, which can be difficult to digest. Breaking these into shorter, more focused sentences would improve clarity.

For example:

Lines 137 – 139:

Original: "Although its focus on possible adverse effects (i.e., fertility issues, irreversible results of gender-affirming hormone treatment) and promoting caution (i.e., in the pre-puberty stage, gender-adverse behaviors may not be due to gender dysphoria, puberty blockers can only be administrated at puberty onset) is understandable, this conventional approach might have severe unintended consequences (Vandermorris & Metzger, 2023)."

Revised: "The focus on possible adverse effects, such as fertility issues and irreversible results of gender-affirming hormone treatment, and promoting caution is understandable. This includes considering that gender-adverse behaviors in the pre-puberty stage may not be due to gender dysphoria, and puberty blockers can only be administered at puberty onset. However, this conventional approach might have severe unintended consequences (Vandermorris & Metzger, 2023)."

OR

Lines 112-114:

Original: "The existing literature underlines the importance of gender expression in the lives of transgender youth and its influence on the development of their gender identity and overall psychological health (Pullen Sansfaçon et al., 2020)."

Revised: "Existing literature highlights the importance of gender expression in the lives of transgender youth. It influences the development of their gender identity and overall psychological health (Pullen Sansfaçon et al., 2020)."

Reply 3: Thank you. Sentences are now simplified or reformulated.

Comment 4:

Conclusions

The conclusions effectively summarize the key points discussed in the paper, reinforcing the importance of the issues raised. They emphasize the need for targeted interventions and evidence-based care, advocating for practical changes in the healthcare system.

They advocate for an inclusive and supportive environment for transgender youth, highlighting the importance of reducing discrimination and stigmatization.

Suggestions for improvement:

Phrases like "potentially enhancing the QoL for the vulnerable transgender youth population" can be more assertive to strengthen the conclusion.

Reply 4: Thank you for the suggestion. We would not like to impose our point of view here, therefore, we decided to keep the phrase as it was.

Comment 5:

Providing a few more concrete steps or examples of how the suggested solutions can be implemented would strengthen the conclusion.

Reply 5: Thank you for your suggestion. In the revised paper, we emphasized the need youth-center approach while assessing the best possible treatment. Based on this, we believe practitioners can implement different solutions taking into account the specificity of their work.

Thank you for your time and helpful recommendations.

Reviewer 2 Report

Comments and Suggestions for Authors

Comments on the Quality of English Language

Author Response

 Dear Editors and Reviewers,

We would like to thank the editors and the reviewers for their positive and encouraging feedback on our submission. The constructive comments of reviewers helped us to significantly improve the quality of our submission. We have been through all comments one by one, edited the manuscript in detail, and added new material where required. We hope the editor and reviewers find the revised version of the manuscript clear and suitable for publication in Healthcare. All changes made in the manuscript are highlighted in red. 

Please find a PDF file with our responses.

Reviewer 3 Report

Comments and Suggestions for Authors

I have reviewed "the opinion" entitled "The Effects of Gender-Affirming Hormone Therapy on Quality of Life: The Importance of Research on Youth." I can say that the subject of “the opinion” the authors discuss is very important. I congratulate the authors for exploring a topic with so many facets and challenges. Below are some points that I think could improve the article.

-         - In some places in the text, the terms "transgender," "gender dysphoria," and "gender incongruence" are used. To help readers follow the subject more easily, I recommend using a single term for concepts that have the same meaning.

-        - I recommend authors use an abbreviation (such as GAHT) for the phrase "gender-affirming hormone therapy" other than its first occurrence.

-          - Only ICD-11 is mentioned in the introduction to the text. It would be appropriate to also mention another current classification system, the DSM-5-TR.

-        -  Highlighting the ethical issues is crucial for a comprehensive understanding of the subject and for fostering a respectful and inclusive approach to transgender healthcare. In this context, ethical issues surrounding the treatment and recognition of transgender individuals should be more prominently featured in the Introduction.

-         - What do the authors think are the possible reasons for the significant increase in the number of young people who identify as transgender in the United States (and likely in many other countries) in recent years? I think the importance of investigating the reasons for this increase should be emphasized in Conclusion.

-         - If they intend to emphasize the lack of knowledge about how medical interventions (e.g. hormone use) alter adolescents' developing mental /psychological state, they should clarify the meaning of “mental well-being”. (I would say that this issue is very important.)

Author Response

Dear Editors and Reviewers,

We would like to thank the editors and the reviewers for their positive and encouraging feedback on our submission. The constructive comments of reviewers helped us to significantly improve the quality of our submission. We have been through all comments one by one, edited the manuscript in detail, and added new material where required. We hope the editor and reviewers find the revised version of the manuscript clear and suitable for publication in Healthcare. All changes made in the manuscript are highlighted in red.

Comment 1:

I have reviewed "the opinion" entitled "The Effects of Gender-Affirming Hormone Therapy on Quality of Life: The Importance of Research on Youth." I can say that the subject of “the opinion” the authors discuss is very important. I congratulate the authors for exploring a topic with so many facets and challenges. Below are some points that I think could improve the article.

In some places in the text, the terms "transgender," "gender dysphoria," and "gender incongruence" are used. To help readers follow the subject more easily, I recommend using a single term for concepts that have the same meaning.

Reply 1: Thank you very much for your suggestion. While it was not possible for us to use the above-mentioned therms as synonyms, we clarified their meanings in the paper.

Comment 2:

I recommend authors use an abbreviation (such as GAHT) for the phrase "gender-affirming hormone therapy" other than its first occurrence.
Reply 2: Thank you. The abbreviation GAHT is now added.

Comment 3:

Only ICD-11 is mentioned in the introduction to the text. It would be appropriate to also mention another current classification system, the DSM-5-TR.
Reply 3: DSM-5-TR is now added alongside IDC-11.

Comment 4:

Highlighting the ethical issues is crucial for a comprehensive understanding of the subject and for fostering a respectful and inclusive approach to transgender healthcare. In this context, ethical issues surrounding the treatment and recognition of transgender individuals should be more prominently featured in the Introduction.
Reply 4: Thank you for this observation. We now added a paragraph related to the Ethical concerns into the body. Furthermore ethical concerns are now mentioned in the introduction.

Comment 5:

What do the authors think are the possible reasons for the significant increase in the number of young people who identify as transgender in the United States (and likely in many other countries) in recent years? I think the importance of investigating the reasons for this increase should be emphasized in Conclusion.
Reply 5: Further details related to this comment are now present in the paper.

Comment 6:
If they intend to emphasize the lack of knowledge about how medical interventions (e.g. hormone use) alter adolescents' developing mental /psychological state, they should clarify the meaning of “mental well-being”. (I would say that this issue is very important.)
Reply 6: Thank you for your suggestion. We now mention well-being in context of aggravating gender-dysphoria.

Thank you for your time and helpful recommendations.

Reviewer 4 Report

Comments and Suggestions for Authors

Introduction: 

1.     For a more affirmative introduction, I suggest not using the term gender incongruence since it sounds pathological. As a non-binary researcher, I will start with the definition of trans and non-binary identities, defining them in an affirmative way as identities that may not aligned with the sex assigned at birth. You can find more affirmative definitions at the American Psychological Association or the WPATH.   

2.   “until January 2022, gender incongruence has been considered a gender identity disorder”. I suggest clarifying in an affirmative way that they have no evidence, and it was completely bias.  

3.   “To align with a modern understanding…” I suggest a more affirmative way such as, to align with evidence and research… modern sound as it is a fashion identity. 

4.    Lines 59-71 needs a little reorganization. Too many ideas paragraph. Maybe enumerating can help to organize and understand better the idea that wants to be transmitted.

5.    “Ethical concerns” are not defined or clarified. 

6.    Benefits over concerns are not mentioned. 

References 

7.    There is a lack of articles from psychological journals which can give so much evidence to the concerns. 

Comments on the Quality of English Language

No comment. 

Author Response

Dear Editors and Reviewers,

We would like to thank the editors and the reviewers for their positive and encouraging feedback on our submission. The constructive comments of reviewers helped us to significantly improve the quality of our submission. We have been through all comments one by one, edited the manuscript in detail, and added new material where required. We hope the editor and reviewers find the revised version of the manuscript clear and suitable for publication in Healthcare. All changes made in the manuscript are highlighted in red.

Comment 1:
For a more affirmative introduction, I suggest not using the term gender incongruence since it sounds pathological. As a non-binary researcher, I will start with the definition of trans and non-binary identities, defining them in an affirmative way as identities that may not aligned with the sex assigned at birth. You can find more affirmative definitions at the American Psychological Association or the WPATH.
Reply 1: The definition is now amended as suggested.

Comment 2:
“until January 2022, gender incongruence has been considered a gender identity disorder”. I suggest clarifying in an affirmative way that they have no evidence, and it was completely bias.
Reply 2: The paragraph is now reformulated, and the statement clarified.

Comment 3:
“To align with a modern understanding…” I suggest a more affirmative way such as, to align with evidence and research… modern sound as it is a fashion identity.
Reply 3: The statement is now reformulated.

Comment 4:
Lines 59-71 needs a little reorganization. Too many ideas paragraph. Maybe enumerating can help to organize and understand better the idea that wants to be transmitted.
Reply 4: The ideas are now reorganized for more clarity.

Comment 5:
“Ethical concerns” are not defined or clarified.
Reply 5: A paragraph focused on ethical concerns is now present in the paper.

Comment 6:
Benefits over concerns are not mentioned.
Reply 6: We now mention benefits, and the eventual limitations of the available treatments and approaches.

Comment 7:
References

There is a lack of articles from psychological journals which can give so much evidence to the concerns.
Reply 7: We have now elaborated the concerns based on additional references.

Thank you for your time and helpful recommendations.

Round 2

Reviewer 2 Report

Comments and Suggestions for Authors

No additional comments. Authors have done an excellent job responding to reviewer feedback and the manuscript is significantly strengthened. 

Reviewer 4 Report

Comments and Suggestions for Authors

The manuscript has been improved.